# A globally convergent fast iterative shrinkage-thresholding algorithm with a new momentum factor for single and multi-objective convex optimization

## Abstract

Convex-composite optimization, which minimizes an objective function represented by the sum of a differentiable function and a convex one, is widely used in machine learning and signal/image processing. Fast Iterative Shrinkage Thresholding Algorithm (FISTA) is a typical method for solving this problem and has a global convergence rate of $O(1/k^2)$. Recently, this has been extended to multi-objective optimization, together with the proof of the $O(1/k^2)$ global convergence rate. However, its momentum factor is classical, and the convergence of its iterates has not been proven. In this work, introducing some additional hyperparameters $(a, b)$, we propose another accelerated proximal gradient method with a general momentum factor, which is new even for the single-objective cases. We show that our proposed method also has a global convergence rate of $O(1/k^2)$ for any $(a, b)$, and further that the generated sequence of iterates converges to a weak Pareto solution when $a$ is positive, an essential property for the finite-time manifold identification. Moreover, we report numerical results with various $(a, b)$, showing that some of these choices give better results than the classical momentum factors.

## 1 Introduction

We consider the following convex-composite single ($m = 1$) or multi-objective ($m \geq 2$) optimization problem:

$$\begin{aligned}
\text{minimize} \quad & F(x) \\
\text{subject to} \quad & x \in \mathbf{R}^n,
\end{aligned} \tag{1}$$

where $F \colon \mathbf{R}^n \to (\mathbf{R} \cup \{\infty\})^m$ is a vector-valued function with $F \coloneqq (F_1, \ldots, F_m)^\top$. For simplicity, we write (1) in this paper as

$$\min_{x \in \mathbf{R}^n} \quad F(x).$$

Hereinafter, we assume the following properties.

**Assumption 1.1**
*Each component $F_i \colon \mathbf{R}^n \to \mathbf{R} \cup \{\infty\}$ is given by*

$$F_i(x) \coloneqq f_i(x) + g_i(x) \quad \text{for all } i = 1, \ldots, m$$

*with convex and continuously differentiable functions $f_i \colon \mathbf{R}^n \to \mathbf{R}, i = 1, \ldots, m$ and closed, proper and convex functions $g_i \colon \mathbf{R}^n \to \mathbf{R} \cup \{\infty\}, i = 1, \ldots, m$, and each $\nabla f_i$ is Lipschitz continuous.*

As suggested in Tanabe et al. (2019), this problem involves many important classes. For example, it can express a convex-constrained problem if each $g_i$ is the indicator function of a convex set $S$, i.e.,

$$\chi_S(x) \coloneqq \begin{cases} 0, & \text{if } x \in S, \\ \infty, & \text{otherwise.} \end{cases} \tag{2}$$

Multi-objective optimization Miettinen (1998) has many applications in engineering Eschenauer et al. (1990), statistics Carrizosa & Frenk (1998), and machine learning (particularly multi-task learning Sener (2018); Lin et al. (2019), neural architecture search Kim et al. (2017); Dong et al. (2018); Elsken et al. (2019), and the accuracy-fairness trade-offs Liu & Vicente (2022)). In the multi-objective case, no single point minimizes all objective functions simultaneously in general. Therefore, we use the concept of *Pareto optimality*. We call a point weakly Pareto optimal if there is no other point where the objective function values are strictly smaller. This generalizes the usual optimality for single-objective problems. In other words, single-objective problems are considered to be included in multi-objective ones. Hence, in the following, unless otherwise noted, we refer to (1) as multi-objective, including the case where $m = 1$.

One of the main strategies for multi-objective problems is the *scalarization approach* Gass & Saaty (1955); Geoffrion (1968); Zadeh (1963), which reduces the original multi-objective problem into a parameterized (or weighted) scalar-valued problem. However, it requires an *a priori* parameters (or weights) selection, which might be challenging. In fact, an example of convex bicriteria problem is provided in (Fliege et al., 2009, Section 7), where almost all choices of parameters fail, leading to unbounded scalarized problems. The meta-heuristics Gandibleux et al. (2004) is also popular, but it has no theoretical convergence properties under reasonable assumptions.

Many descent methods have been developed in recent years Fukuda & Graña Drummond (2014), overcoming those drawbacks. They decrease one or more objectives at each iteration or within some iterations, and their global convergence property can be analyzed under reasonable assumptions. This type of method is suitable for improving tentative solutions, since it always generates a solution that improves the starting point. Empirically, it is also known to obtain a variety of Pareto solutions by starting from different initial points. For example, the steepest descent method Fliege & Svaiter (2000); Fliege et al. (2019); Désidéri (2012) converges globally to Pareto solutions for differentiable multi-objective problems. From a practical point of view, its applicability has also been reported in multi-task learning Sener (2018); Lin et al. (2019). Afterwards, the projected gradient Fukuda & Graña Drummond (2013), Newton's Fliege et al. (2009); Gonçalves et al. (2022), trust-region Carrizo et al. (2016), and conjugate gradient methods Lucambio Pérez & Prudente (2018) were also considered. Moreover, the proximal point Bonnel et al. (2005) and the inertial forward-backward methods Boţ & Grad (2018) can solve infinite-dimensional vector optimization problems.

For (1), the proximal gradient method Tanabe et al. (2019; 2023a) is effective. Using it, the merit function Tanabe et al. (2023c), which returns zero at the Pareto solutions and strictly positive values otherwise, converges to zero with rate $O(1/k)$ under reasonable assumptions. It is also shown that the generated sequence of iterates converges to a weak Pareto solution Bello Cruz et al. (2022). On the other hand, the accelerated proximal gradient method Tanabe et al. (2023b), which generalizes the Fast Iterative Shrinkage Thresholding Algorithm (FISTA) Beck & Teboulle (2009) for convex-composite single-objective problems to multi-objective optimization, has also been considered, along with a proof of the merit function's $O(1/k^2)$ convergence rate. However, the momentum factor used there is classical ($t_1 = 1, t_{k+1} = \sqrt{t_k^2 + 1/4} + 1/2$), and the iterates' convergence is not proven.

This paper generalizes the associated factor by $t_1 = 1, t_{k+1} = \sqrt{t_k^2 - at_k + b} + 1/2$ with hyperparameters $a \in [0, 1), b \in [a^2/4, 1/4]$. This is new even in the single-objective context, and it generalizes well-known factors. For example, when $a = 0$ and $b = 1/4$, it reduces to $t_1 = 1, t_{k+1} = \sqrt{t_k^2 + 1/4} + 1/2$, proposed in Nesterov (1983); Beck & Teboulle (2009), and when $b = a^2/4$, it gives $t_k = (1 - a)k/2 + (1 + a)/2$, suggested in Chambolle & Dossal (2015); Attouch & Peypouquet (2016); Attouch et al. (2018); Su et al. (2016). We show that the merit function converges to zero with rate $O(1/k^2)$ for any $(a, b)$. In addition, we prove the iterates' convergence to a weak Pareto solution when $a > 0$. While the generalization of the momentum factor is an important aspect of our work, it is crucial to emphasize that our primary contribution lies in addressing the challenge of ensuring convergence in accelerated gradient methods for multi-objective optimization. As discussed in Section 4, this suggests that the proposed method might achieve finite-iteration manifold (active set) identification Sun et al. (2019) without the assumption of strong convexity.

Furthermore, we carry out numerical experiments with various $(a, b)$ and observe that some $(a, b)$ yield better results than the classical factors. However, it is important to note that the primary focus and significance of

our work lies in the theoretical advancement of convergence assurance rather than in the specific properties of the new momentum factor.

The outline of this paper is as follows. We present some notations and definitions used in this paper in Section 2.1. Section 2.2 recalls the accelerated proximal gradient method for (1) and its associated results. We generalize the momentum factor and prove that it preserves an $O(1/k^2)$ convergence rate in Section 3, and we demonstrate the convergence of the iterates in Section 4. Finally, Section 5 provides numerical experiments and compares the numerical performances depending on the hyperparameters.

## 2 Preliminaries

### 2.1 Definitions and notations

For every natural number $d$, write the $d$-dimensional real space by $\mathbf{R}^d$, and define

$$\mathbf{R}_+^d := \left\{ v \in \mathbf{R}^d \mid v_i \geq 0, i = 1, \ldots, d \right\}.$$

This induces the partial orders: for any $v^1, v^2 \in \mathbf{R}^d$, $v^1 \leq v^2$ (alternatively, $v^2 \geq v^1$) if $v^2 - v^1 \in \mathbf{R}_+^d$ and $v^1 < v^2$ (alternatively, $v^2 > v^1$) if $v^2 - v^1 \in \text{int} \, \mathbf{R}_+^d$. In other words, $v^1 \leq v^2$ and $v^1 < v^2$ mean that $v_i^1 \leq v_i^2$ and $v_i^1 < v_i^2$ for all $i = 1, \ldots, d$, respectively. Furthermore, let $\langle \cdot, \cdot \rangle$ be the Euclidean inner product in $\mathbf{R}^d$, i.e., $\langle v^1, v^2 \rangle := \sum_{i=1}^d v_i^1 v_i^2$, and let $\|\cdot\|$ be the Euclidean norm, i.e., $\|v\| := \sqrt{\langle v, v \rangle}$. Moreover, we define the $\ell_1$-norm and $\ell_\infty$-norm by $\|v\|_1 := \sum_{i=1}^m |v_i|$ and $\|v\|_\infty := \max_{i=1,\ldots,d} |v_i|$, respectively.

We introduce some concepts used in the problem (1). Recall that $x^* \in \mathbf{R}^n$ is *Pareto optimal* if there is no $x \in \mathbf{R}^n$ such that $F(x) \leq F(x^*)$ and $F(x) \neq F(x^*)$. Likewise,

$$X^* := \{x^* \in \mathbf{R}^n \mid \text{There does not exist } x \in \mathbf{R}^n \text{ such that } F(x) < F(x^*)\} \tag{3}$$

is the set of *weakly Pareto optimal* solutions for (1). When $m = 1$, $X^*$ reduces to the optimal solution set. It is known that all Pareto optimal points are weakly Pareto optimal, and the converse is true if every objective function is strictly convex. Moreover, define the effective domain of $F$ by

$$\text{dom} \, F := \{x \in \mathbf{R}^n \mid F(x) < \infty\},$$

and write the level set of $F$ on $c \in \mathbf{R}^m$ as

$$\mathcal{L}_F(c) := \{x \in \mathbf{R}^n \mid F(x) \leq c\}. \tag{4}$$

Furthermore, we express the image of $A \subseteq \mathbf{R}^n$ and the inverse image of $B \subseteq (\mathbf{R} \cup \{\infty\})^m$ under $F$ as

$$F(A) := \{F(x) \in \mathbf{R}^m \mid x \in A\} \quad \text{and} \quad F^{-1}(B) := \{x \in \mathbf{R}^n \mid F(x) \in B\},$$

respectively.

Finally, let us recall the merit function $u_0 \colon \mathbf{R}^n \to \mathbf{R} \cup \{\infty\}$ for (1) proposed in Tanabe et al. (2023c):

$$u_0(x) := \sup_{z \in \mathbf{R}^n} \min_{i=1,\ldots,m} [F_i(x) - F_i(z)], \tag{5}$$

which returns zero at optimal solutions and strictly positive values otherwise. Because of its many desirable properties proved in Tanabe et al. (2023c), we can think of $u_0(x)$ representing, in a sense, how far $x$ is from the Pareto solution. For example, the following theorem shows that $u_0$ is a merit function in the Pareto sense.

**Theorem 2.1**

*(Tanabe et al., 2023c, Theorem 3.1) Let $u_0$ be defined by (5). Then, $u_0(x) \geq 0$ for all $x \in \mathbf{R}^n$. Moreover, $x \in \mathbf{R}^n$ is weakly Pareto optimal for (1) if and only if $u_0(x) = 0$.*

Note that when $m = 1$, we have

$$u_0(x) = F_1(x) - F_1^*,$$

where $F_1^*$ is the optimal objective value. Clearly, this is a merit function for scalar-valued optimization.

## 2.2 The accelerated proximal gradient method for multi-objective optimization

This subsection recalls the accelerated proximal gradient method for (1) proposed in Tanabe et al. (2023b) and its main results. Recall that each $F_i$ is the sum of a continuously differentiable function $f_i$ and a closed, proper, and convex function $g_i$, and that $\nabla f_i$ is Lipschitz continuous with Lipschitz constant $L_i > 0$. Define

$$L := \max_{i=1,\dots,m} L_i.$$

The method solves the following subproblem at each iteration for given $x \in \operatorname{dom} F$, $y \in \mathbf{R}^n$, and $\ell \geq L$:

$$\min_{z \in \mathbf{R}^n} \quad \varphi_\ell^{\mathrm{acc}}(z; x, y), \tag{6}$$

where

$$\varphi_\ell^{\mathrm{acc}}(z; x, y) := \max_{i=1,\dots,m} [\langle \nabla f_i(y), z - y \rangle + g_i(z) + f_i(y) - F_i(x)] + \frac{\ell}{2} \|z - y\|^2.$$

Note that if $f_i = 0$ for every $i$ and $\ell = 0$, then we have $u_0(x) = -\min_x \varphi_\ell^{\mathrm{acc}}(z; x, y)$. From the strong convexity, (6) has a unique optimal solution $p_\ell^{\mathrm{acc}}(x, y)$, i.e.,

$$p_\ell^{\mathrm{acc}}(x, y) := \operatorname*{argmin}_{z \in \mathbf{R}^n} \varphi_\ell^{\mathrm{acc}}(z; x, y). \tag{7}$$

The following proposition characterizes weak Pareto optimality in terms of the mapping $p_\ell^{\mathrm{acc}}$.

**Proposition 2.1**

*(Tanabe et al., 2023b, Proposition 4.1) Let $p_\ell^{\mathrm{acc}}(x, y)$ be defined by (7). Then, the following statements hold.*

(i) *$y \in \mathbf{R}^n$ is weakly Pareto optimal for (1) if and only if $p_\ell^{\mathrm{acc}}(x, y) = y$ for some $x \in \mathbf{R}^n$.*

(ii) *$p_\ell^{\mathrm{acc}}$ is locally Hölder continuous with exponent $1/2$, i.e., for any bounded set $W \subseteq \mathbf{R}^n$, there exists $M > 0$ such that*

$$\|p_\ell^{\mathrm{acc}}(\hat{x}, \hat{y}) - p_\ell^{\mathrm{acc}}(\check{x}, \check{y})\| \leq M \|(\hat{x}, \hat{y}) - (\check{x}, \check{y})\|^{1/2}$$

*for all $\hat{x}, \hat{y}, \check{x}, \check{y} \in W$.*

This implies that using $\|p_\ell^{\mathrm{acc}}(x, y) - y\|_\infty < \varepsilon$ for some $\varepsilon > 0$ is reasonable as the stopping criteria. We state below the accelerated proximal gradient method for (1).

---

**Algorithm 1** Accelerated proximal gradient method for (1)

---

**Input:** Set $x^0 = y^1 \in \operatorname{dom} F, \ell \geq L, \varepsilon > 0$.
**Output:** $x^k \in \operatorname{dom} F$: An approximate weak Pareto optimal solution
 1: $k \leftarrow 0$
 2: $t_1 \leftarrow 1$
 3: **repeat**
 4:     $k \leftarrow k + 1$
 5:     $x^k \leftarrow p_\ell^{\mathrm{acc}}(x^{k-1}, y^k)$
 6:     $t_{k+1} \leftarrow \sqrt{t_k^2 + 1/4} + 1/2$
 7:     $\gamma_k \leftarrow (t_k - 1)/t_{k+1}$
 8:     $y^{k+1} \leftarrow x^k + \gamma_k(x^k - x^{k-1})$
 9: **until** $\|x^k - y^k\|_\infty < \varepsilon$

---

Algorithm 1 generates $\{x^k\}$ such that $\{u_0(x^k)\}$ converges to zero with rate $O(1/k^2)$ under the following assumption. This assumption is also used to analyze the proximal gradient method without acceleration Tanabe et al. (2023a) and is not particularly strong as suggested in (Tanabe et al., 2023a, Remark 5.3); it is satisfied for level bounded functions such as $\ell_1$-norm, for example.

**Assumption 2.1**
*(Tanabe et al., 2023a, Assumption 5.1) Let $X^*$ and $\mathcal{L}_F$ be defined by (3) and (4), respectively. Then, for all $x \in \mathcal{L}_F(F(x^0))$, there exists $x^* \in X^*$ such that $F(x^*) \leq F(x)$ and*

$$R := \sup_{F^* \in F(X^* \cap \mathcal{L}_F(F(x^0)))} \inf_{z \in F^{-1}(\{F^*\})} \left\| z - x^0 \right\|^2 < \infty. \tag{8}$$

**Theorem 2.2**
*(Tanabe et al., 2023b, Theorem 5.2) Under Assumptions 1.1 and 2.1, Algorithm 1 generates $\{x^k\}$ such that*

$$u_0(x^k) \leq \frac{2\ell R}{(k+1)^2} \quad \text{for all } k \geq 1,$$

*where $R \geq 0$ is given by (8), and $u_0$ is a merit function defined by (5).*

The following corollary shows the global convergence of Algorithm 1.

**Corollary 2.1**
*(Tanabe et al., 2023b, Corollary 5.2) Suppose that Assumptions 1.1 and 2.1 hold. Then, every accumulation point of $\{x^k\}$ generated by Algorithm 1 is weakly Pareto optimal for (1).*

## 3 Generalization of the momentum factor and convergence rate analysis

This section generalizes the momentum factor $\{t_k\}$ used in Algorithm 1 and shows that the $O(1/k^2)$ convergence rate also holds in that case. First, we describe below the algorithm in which we replace line 6 of Algorithm 1 by a formula using given constants $a \in [0, 1)$ and $b \in [a^2/4, 1/4]$:

---

**Algorithm 2** Accelerated proximal gradient method with general stepsizes for (1)

---

**Input:** Set $x^0 = y^1 \in \text{dom } F, \ell \geq L, \varepsilon > 0, a \in [0, 1), b \in [a^2/4, 1/4]$.
**Output:** $x^k \in \text{dom } F$: An approximate weak Pareto optimal solution
1: $k \leftarrow 0$
2: $t_1 \leftarrow 1$
3: **repeat**
4:     $k \leftarrow k + 1$
5:     $x^k \leftarrow p_\ell^{\text{acc}}(x^{k-1}, y^k)$
6:     $t_{k+1} \leftarrow \sqrt{t_k^2 - at_k + b} + 1/2$
7:     $\gamma_k \leftarrow (t_k - 1)/t_{k+1}$
8:     $y^{k+1} \leftarrow x^k + \gamma_k(x^k - x^{k-1})$
9: **until** $\left\| x^k - y^k \right\|_\infty < \varepsilon$

---

The sequence $\{t_k\}$ defined in lines 2 and 6 of Algorithm 2 generalizes the well-known momentum factors in single-objective accelerated methods. For example, when $a = 0$ and $b = 1/4$, they coincide with the one in Algorithm 1 and the original FISTA Nesterov (1983); Beck & Teboulle (2009) ($t_1 = 1$ and $t_{k+1} = (1 + \sqrt{1 + 4t_k^2})/2$). Moreover, if $b = a^2/4$, then $\{t_k\}$ has the general term $t_k = (1-a)k/2 + (1+a)/2$, which corresponds to the one used in Chambolle & Dossal (2015); Su et al. (2016); Attouch & Peypouquet (2016); Attouch et al. (2018). This means that our generalization allows a finer tuning of the algorithm by varying $a$ and $b$.

We present below the main theorem of this section.

**Theorem 3.1**
*Let $\{x^k\}$ be a sequence generated by Algorithm 2 and recall that $u_0$ is given by (5). Then, under Assumption 1.1, the following two equations hold:*

*(i) $F_i(x^k) \leq F_i(x^0)$ for all $i = 1, \ldots, m$ and $k \geq 0$;*

(ii) $u_0(x^k) = O(1/k^2)$ as $k \to \infty$ under *Assumption 2.1*.

Claim (i) means that $\{x^k\} \subseteq \mathcal{L}_F(F(x^0))$, where $\mathcal{L}_F$ denotes the level set of $F$ (cf. (4)). Note, however, that the objective functions are generally not monotonically non-increasing. Claim (ii) also claims the global convergence rate.

Before proving Theorem 3.1, let us give several lemmas. First, we present some properties of $\{t_k\}$ and $\{\gamma_k\}$.

**Lemma 3.1**
*Let $\{t_k\}$ and $\{\gamma_k\}$ be defined by lines 2, 6 and 7 in Algorithm 2 for arbitrary $a \in [0, 1)$ and $b \in [a^2/4, 1/4]$. Then, the following inequalities hold for all $k \geq 1$.*

(i) $t_{k+1} \geq t_k + \dfrac{1-a}{2}$ and $t_k \geq \dfrac{1-a}{2}k + \dfrac{1+a}{2}$;

(ii) $t_{k+1} \leq t_k + \dfrac{1-a+\sqrt{4b-a^2}}{2}$ and $t_k \leq \dfrac{1-a+\sqrt{4b-a^2}}{2}(k-1) + 1 \leq k$;

(iii) $t_k^2 - t_{k+1}^2 + t_{k+1} = at_k - b + \dfrac{1}{4} \geq at_k$;

(iv) $0 \leq \gamma_k \leq \dfrac{k-1}{k+1/2}$;

(v) $1 - \gamma_k^2 \geq \dfrac{1}{t_k}$.

*Proof.* Claim (i): From the definition of $\{t_k\}$, we have

$$t_{k+1} = \sqrt{t_k^2 - at_k + b} + \frac{1}{2} = \sqrt{\left(t_k - \frac{a}{2}\right)^2 + \left(b - \frac{a^2}{4}\right)} + \frac{1}{2}. \tag{9}$$

Since $b \geq a^2/4$, we get

$$t_{k+1} \geq \left|t_k - \frac{a}{2}\right| + \frac{1}{2}.$$

Since $t_1 = 1 \geq a/2$, we can quickly see that $t_k \geq a/2$ for any $k$ by induction. Thus, we have

$$t_{k+1} \geq t_k + \frac{1-a}{2}.$$

Applying the above inequality recursively, we obtain

$$t_k \geq \frac{1-a}{2}(k-1) + t_1 = \frac{1-a}{2}k + \frac{1+a}{2}.$$

Claim (ii): From (9) and the relation $\sqrt{\alpha + \beta} \leq \sqrt{\alpha} + \sqrt{\beta}$ with $\alpha, \beta \geq 0$, we get the first inequality. Using it recursively, it follows that

$$t_k \leq \frac{1-a+\sqrt{4b-a^2}}{2}(k-1) + t_1 = \frac{1-a+\sqrt{4b-a^2}}{2}(k-1) + 1.$$

Since $a \in [0, 1), b \in [a^2/4, 1/4]$, we observe that

$$\frac{1-a+\sqrt{4b-a^2}}{2} \leq \frac{1-a+\sqrt{1-a^2}}{2} \leq 1.$$

Hence, the above two inequalities lead to the desired result.

Claim (iii): An easy computation shows that

$$t_k^2 - t_{k+1}^2 + t_{k+1} = t_k^2 - \left[\sqrt{t_k^2 - at_k + b} + \frac{1}{2}\right]^2 + \sqrt{t_k^2 - at_k + b} + \frac{1}{2}$$

$$= at_k - b + \frac{1}{4} \geq at_k,$$

where the inequality holds since $b \leq 1/4$.

Claim (iv): The first inequality is clear from the definition of $\gamma_k$ since claim (i) yields $t_k \geq 1$. Again, the definition of $\gamma_k$ and claim (i) give

$$\gamma_k = \frac{t_k - 1}{t_{k+1}} \leq \frac{t_k - 1}{t_k + (1-a)/2} = 1 - \frac{3-a}{2t_k + 1 - a}.$$

Combining with claim (ii), we get

$$\gamma_k \leq 1 - \frac{3-a}{\left(1 - a + \sqrt{4b - a^2}\right)(k-1) + 3 - a}$$

$$= \frac{\left(1 - a + \sqrt{4b - a^2}\right)(k-1)}{\left(1 - a + \sqrt{4b - a^2}\right)(k-1) + 3 - a} \qquad (10)$$

$$= \frac{k-1}{k - 1 + (3-a)/\left(1 - a + \sqrt{4b - a^2}\right)}.$$

On the other hand, it follows that

$$\min_{a \in [0,1), b \in [a^2/4, 1/4]} \frac{3-a}{1 - a + \sqrt{4b - a^2}} = \min_{a \in [0,1)} \frac{3-a}{1 - a + \sqrt{1-a^2}} = \frac{3}{2}, \qquad (11)$$

where the second equality follows from the monotonic non-decreasing property implied by

$$\frac{\mathrm{d}}{\mathrm{d}a}\left(\frac{3-a}{1 - a + \sqrt{1-a^2}}\right) = \frac{2\sqrt{1-a^2} + 3a - 1}{\left(\sqrt{1-a^2} - a + 1\right)^2 \sqrt{1-a^2}} > 0 \quad \text{for all } a \in [0,1).$$

Combining (10) and (11), we obtain $\gamma_k \leq (k-1)/(k+1/2)$.

Claim (v): claim (i) implies that $t_{k+1} > t_k \geq 1$. Thus, the definition of $\gamma_k$ implies that

$$1 - \gamma_k^2 = 1 - \left(\frac{t_k - 1}{t_{k+1}}\right)^2 \geq 1 - \left(\frac{t_k - 1}{t_k}\right)^2 = \frac{2t_k - 1}{t_k^2} \geq \frac{2t_k - t_k}{t_k^2} = \frac{1}{t_k}.$$

$\square$

As in Tanabe et al. (2023b), we also introduce $\sigma_k \colon \mathbf{R}^n \to \mathbf{R} \cup \{-\infty\}$ and $\rho_k \colon \mathbf{R}^n \to \mathbf{R}$ for $k \geq 0$ as follows, which assist the analysis:

$$\sigma_k(z) := \min_{i=1,\ldots,m} \left[F_i(x^k) - F_i(z)\right],$$

$$\rho_k(z) := \left\|t_{k+1}x^{k+1} - (t_{k+1} - 1)x^k - z\right\|^2. \qquad (12)$$

The following lemma on $\sigma_k$ is helpful in the subsequent discussions.

**Lemma 3.2**

*(Tanabe et al., 2023b, Lemma 5.1) Let $\{x^k\}$ and $\{y^k\}$ be sequences generated by Algorithm 2. Then, under Assumption 1.1, the following inequalities hold for all $z \in \mathbf{R}^n$ and $k \geq 0$:*

*(i)* $\sigma_{k+1}(z) \leq -\frac{\ell}{2}\left(2\langle x^{k+1} - y^{k+1}, y^{k+1} - z\rangle + \left\|x^{k+1} - y^{k+1}\right\|^2\right) - \frac{\ell - L}{2}\left\|x^{k+1} - y^{k+1}\right\|^2;$

*(ii)* $\sigma_k(z) - \sigma_{k+1}(z) \geq \dfrac{\ell}{2}\left(2\langle x^{k+1} - y^{k+1}, y^{k+1} - x^k\rangle + \left\|x^{k+1} - y^{k+1}\right\|^2\right) + \dfrac{\ell - L}{2}\left\|x^{k+1} - y^{k+1}\right\|^2.$

Therefore, from Lemma 3.1 (v), we can obtain the following result quickly in the same way as in the proof of (Tanabe et al., 2023b, Corollary 5.1).

**Lemma 3.3**
*Let $\{x^k\}$ and $\{y^k\}$ be sequences generated by Algorithm 2. Then, under Assumption 1.1, we have*

$$\sigma_{k_1}(z) - \sigma_{k_2}(z) \geq \frac{\ell}{2}\left(\left\|x^{k_2} - x^{k_2-1}\right\|^2 - \left\|x^{k_1} - x^{k_1-1}\right\|^2 + \sum_{k=k_1}^{k_2-1}\frac{1}{t_k}\left\|x^k - x^{k-1}\right\|^2\right)$$

*for any $k_2 \geq k_1 \geq 1$.*

We can now show the first part of Theorem 3.1.

*Theorem 3.1 (i).* From Lemma 3.3, we can prove this part with similar arguments used in the proof of (Tanabe et al., 2023b, Theorem 5.1). □

The next step is to prepare the proof of Theorem 3.1 (ii). First, we mention the following relation, used frequently hereafter:

$$\left\|v^2 - v^1\right\|^2 + 2\langle v^2 - v^1, v^1 - v^3\rangle = \left\|v^2 - v^3\right\|^2 - \left\|v^1 - v^3\right\|^2, \tag{13}$$

$$\sum_{s=1}^{r}\sum_{p=1}^{s} A_p = \sum_{p=1}^{r}\sum_{s=p}^{r} A_p \tag{14}$$

for any vectors $v^1, v^2, v^3$ and sequence $\{A_p\}$. With these, we show the lemma below, which is similar to (Tanabe et al., 2023b, Lemma 5.2) but more complex due to the generalization of $\{t_k\}$.

**Lemma 3.4**
*Let $\{x^k\}$ and $\{y^k\}$ be sequences generated by Algorithm 2. Also, let $\sigma_k$ and $\rho_k$ be defined by (12). Then, under Assumption 1.1, we have*

$$\frac{\ell}{2}\left\|x^0 - z\right\|^2$$

$$\geq \frac{1}{1-a}\left[t_{k+1}^2 - at_{k+1} + \left(\frac{1}{4} - b\right)k\right]\sigma_{k+1}(z)$$

$$+ \frac{\ell}{2(1-a)}\left[a(t_{k+1}^2 - t_{k+1}) + \left(\frac{1}{4} - b\right)k\right]\left\|x^{k+1} - x^k\right\|^2$$

$$+ \frac{\ell}{2(1-a)}\sum_{p=1}^{k}\left[a^2(t_p - 1) + \left(\frac{1}{4} - b\right)\frac{p - t_p + a(t_p - 1)}{t_p}\right]\left\|x^p - x^{p-1}\right\|^2$$

$$+ \frac{\ell}{2}\rho_k(z) + \frac{\ell - L}{2}\sum_{p=1}^{k} t_{p+1}^2\left\|x^{p+1} - y^{p+1}\right\|^2$$

*for all $k \geq 0$ and $z \in \mathbf{R}^n$.*

*Proof.* Let $p \geq 1$ and $z \in \mathbf{R}^n$. Recall that Lemma 3.2 gives

$$-\sigma_{p+1}(z) \geq \frac{\ell}{2}\left[2\langle x^{p+1} - y^{p+1}, y^{p+1} - z\rangle + \left\|x^{p+1} - y^{p+1}\right\|^2\right]$$

$$+ \frac{\ell - L}{2}\left\|x^{p+1} - y^{p+1}\right\|^2,$$

$$\sigma_p(z) - \sigma_{p+1}(z) \geq \frac{\ell}{2}\left[2\langle x^{p+1} - y^{p+1}, y^{p+1} - x^p\rangle + \left\|x^{p+1} - y^{p+1}\right\|^2\right]$$

$$+ \frac{\ell - L}{2}\left\|x^{p+1} - y^{p+1}\right\|^2.$$

We then multiply the second inequality above by $(t_{p+1} - 1)$ and add it to the first one:

$$(t_{p+1} - 1)\sigma_p(z) - t_{p+1}\sigma_{p+1}(z)$$
$$\geq \frac{\ell}{2}\left[t_{p+1}\|x^{p+1} - y^{p+1}\|^2 + 2\langle x^{p+1} - y^{p+1}, t_{p+1}y^{p+1} - (t_{p+1} - 1)x^p - z\rangle\right]$$
$$+ \frac{\ell - L}{2}t_{p+1}\|x^{p+1} - y^{p+1}\|^2.$$

Multiplying this inequality by $t_{p+1}$ and using the relation $t_p^2 = t_{p+1}^2 - t_{p+1} + (at_p - b + 1/4)$ (cf. Lemma 3.1 (iii)), we get

$$t_p^2\sigma_p(z) - t_{p+1}^2\sigma_{p+1}(z) \geq \frac{\ell}{2}\Big[\|t_{p+1}(x^{p+1} - y^{p+1})\|^2$$
$$+ 2t_{p+1}\langle x^{p+1} - y^{p+1}, t_{p+1}y^{p+1} - (t_{p+1} - 1)x^p - z\rangle\Big]$$
$$+ \frac{\ell - L}{2}t_{p+1}^2\|x^{p+1} - y^{p+1}\|^2 + \left(at_p - b + \frac{1}{4}\right)\sigma_p(z).$$

Applying (13) to the right-hand side of the last inequality with

$$v^1 := t_{p+1}y^{p+1}, \quad v^2 := t_{p+1}x^{p+1}, \quad v^3 := (t_{p+1} - 1)x^p + z.$$

we get

$$t_p^2\sigma_p(z) - t_{p+1}^2\sigma_{p+1}(z)$$
$$\geq \frac{\ell}{2}\left[\|t_{p+1}x^{p+1} - (t_{p+1} - 1)x^p - z\|^2 - \|t_{p+1}y^{p+1} - (t_{p+1} - 1)x^p - z\|^2\right]$$
$$+ \frac{\ell - L}{2}t_{p+1}^2\|x^{p+1} - y^{p+1}\|^2 + \left(at_p - b + \frac{1}{4}\right)\sigma_p(z).$$

Recall that $\rho_p(z) := \|t_{p+1}x^{p+1} - (t_{p+1} - 1)x^p - z\|^2$. Then, considering the definition of $y^p$ given in line 8 of Algorithm 2, we obtain

$$t_p^2\sigma_p(z) - t_{p+1}^2\sigma_{p+1}(z) \geq \frac{\ell}{2}[\rho_p(z) - \rho_{p-1}(z)] + \frac{\ell - L}{2}t_{p+1}^2\|x^{p+1} - y^{p+1}\|^2 + \left(at_p - b + \frac{1}{4}\right)\sigma_p(z).$$

Now, let $k \geq 0$. Lemma 3.3 with $(k_1, k_2) = (p, k + 1)$ implies

$$t_p^2\sigma_p(z) - t_{p+1}^2\sigma_{p+1}(z) \geq \frac{\ell}{2}[\rho_p(z) - \rho_{p-1}(z)] + \frac{\ell - L}{2}t_{p+1}^2\|x^{p+1} - y^{p+1}\|^2$$
$$+ \left(at_p - b + \frac{1}{4}\right)\left[\sigma_{k+1}(z) + \frac{\ell}{2}\left(\|x^{k+1} - x^k\|^2 - \|x^p - x^{p-1}\|^2 + \sum_{r=p}^{k}\frac{1}{t_r}\|x^r - x^{r-1}\|^2\right)\right].$$

Adding up the above inequality from $p = 1$ to $p = k$, the fact that $t_1 = 1$ and $\rho_0(z) = \|x^1 - z\|^2$ leads to

$$\sigma_1(z) - t_{k+1}^2\sigma_{k+1}(z) \geq \frac{\ell}{2}\left[\rho_k(z) - \|x^1 - z\|^2\right] + \frac{\ell - L}{2}\sum_{p=1}^{k}t_{p+1}^2\|x^{p+1} - y^{p+1}\|^2$$
$$+ \left(a\sum_{p=1}^{k}t_p + \left(\frac{1}{4} - b\right)k\right)\left[\sigma_{k+1}(z) + \frac{\ell}{2}\|x^{k+1} - x^k\|^2\right]$$
$$- \frac{\ell}{2}\sum_{p=1}^{k}\left(at_p - b + \frac{1}{4}\right)\|x^p - x^{p+1}\|^2$$
$$+ \frac{\ell}{2}\sum_{p=1}^{k}\left(at_p - b + \frac{1}{4}\right)\sum_{r=p}^{k}\frac{1}{t_r}\|x^r - x^{r-1}\|^2. \quad (15)$$

Let us write the last two terms of the right-hand side for (15) as $S_1$ and $S_2$, respectively. Equation (14) yields

$$S_2 = \frac{\ell}{2} \sum_{r=1}^{k} \sum_{p=1}^{r} \left( at_p - b + \frac{1}{4} \right) \frac{1}{t_r} \left\| x^r - x^{r-1} \right\|^2$$

$$= \frac{\ell}{2} \sum_{p=1}^{k} \sum_{r=1}^{p} \left( at_r - b + \frac{1}{4} \right) \frac{1}{t_p} \left\| x^p - x^{p-1} \right\|^2.$$

Hence, it follows that

$$S_1 + S_2 = \frac{\ell}{2} \sum_{p=1}^{k} \left[ \frac{1}{t_p} \sum_{r=1}^{p} \left( at_r - b + \frac{1}{4} \right) - \left( at_p - b + \frac{1}{4} \right) \right] \left\| x^p - x^{p-1} \right\|^2$$

$$= \frac{\ell}{2} \sum_{p=1}^{k} \frac{1}{t_p} \left[ a \left( \sum_{r=1}^{p-1} t_r - t_p^2 + t_p \right) + \left( \frac{1}{4} - b \right) (p - t_p) \right] \left\| x^p - x^{p-1} \right\|^2. \quad (16)$$

Again $t_1 = 1$ gives

$$-t_p^2 + t_p = \sum_{r=1}^{p-1} (-t_{r+1}^2 + t_{r+1} + t_r^2 - t_r) = \sum_{r=1}^{p-1} \left( -(1-a)t_r - b + \frac{1}{4} \right)$$

$$= -(1-a) \sum_{r=1}^{p-1} t_r + \left( \frac{1}{4} - b \right) (p-1),$$

where the second equality comes from Lemma 3.1 (iii). Thus, we get

$$\sum_{r=1}^{p-1} t_r = \frac{t_p^2 - t_p}{1-a} + \left( \frac{1}{4} - b \right) \frac{p-1}{1-a}. \quad (17)$$

Substituting this into (16), it follows that

$$S_1 + S_2 = \frac{\ell}{2(1-a)} \sum_{p=1}^{k} \left[ a^2(t_p - 1) + \left( \frac{1}{4} - b \right) \frac{p - t_p + a(t_p - 1)}{t_p} \right] \left\| x^p - x^{p-1} \right\|^2.$$

Combined with (15) and (17), we have

$$\sigma_1(z) - t_{k+1}^2 \sigma_{k+1}(z)$$

$$\geq \frac{\ell}{2} \left[ \rho_k(z) - \left\| x^1 - z \right\|^2 \right] + \frac{\ell - L}{2} \sum_{p=1}^{k} t_{p+1}^2 \left\| x^{k+1} - y^{k+1} \right\|^2$$

$$+ \frac{1}{1-a} \left[ a(t_{k+1}^2 - t_{k+1}) + \left( \frac{1}{4} - b \right) k \right] \left[ \sigma_{k+1}(z) + \frac{\ell}{2} \left\| x^{k+1} - x^k \right\|^2 \right]$$

$$+ \frac{\ell}{2(1-a)} \sum_{p=1}^{k} \left[ a^2(t_p - 1) + \left( \frac{1}{4} - b \right) \frac{p - t_p + a(t_p - 1)}{t_p} \right] \left\| x^p - x^{p-1} \right\|^2.$$

Easy calculations give

$$
\sigma_1(z) + \frac{\ell}{2}\big\|x^1 - z\big\|^2
$$

$$
\begin{aligned}
\geq & \frac{1}{1-a}\left[t_{k+1}^2 - at_{k+1} + \left(\frac{1}{4} - b\right)k\right]\sigma_{k+1}(z) \\
& + \frac{\ell}{2(1-a)}\left[a(t_{k+1}^2 - t_{k+1}) + \left(\frac{1}{4} - b\right)k\right]\big\|x^{k+1} - x^k\big\|^2 \\
& + \frac{\ell}{2(1-a)}\sum_{p=1}^{k}\left[a^2(t_p - 1) + \left(\frac{1}{4} - b\right)\frac{p - t_p + a(t_p - 1)}{t_p}\right]\big\|x^p - x^{p-1}\big\|^2 \\
& + \frac{\ell}{2}\rho_k(z) + \frac{\ell - L}{2}\sum_{p=1}^{k}t_{p+1}^2\big\|x^{k+1} - y^{k+1}\big\|^2.
\end{aligned}
$$

Lemma 3.2 (i) with $k = 0$ and $y^1 = x^0$ and (13) with $(v^1, v^2, v^3) = (x^0, x^1, z)$ lead to

$$
\sigma_1(z) \leq -\frac{\ell}{2}\left[\big\|x^1 - z\big\|^2 - \big\|x^0 - z\big\|^2\right] - \frac{\ell - L}{2}\big\|x^1 - y^1\big\|^2.
$$

From the above two inequalities and the fact that $\ell \geq L$, we can derive the desired inequality. $\qquad\square$

Let us define the linear function $P\colon \mathbf{R} \to \mathbf{R}$ and quadratic ones $Q_1\colon \mathbf{R} \to \mathbf{R}$, $Q_2\colon \mathbf{R} \to \mathbf{R}$, and $Q_3\colon \mathbf{R} \to \mathbf{R}$ by

$$
\begin{aligned}
P(\alpha) &:= \frac{a^2(\alpha - 1)}{2}, \\
Q_1(\alpha) &:= \frac{1-a}{4}\alpha^2 + \left[1 - \frac{a}{2} + \frac{1 - 4b}{4(1-a)}\right]\alpha + 1, \\
Q_2(\alpha) &:= \frac{a(1-a)}{4}\alpha^2 + \left[\frac{a}{2} + \frac{1 - 4b}{4(1-a)}\right]\alpha, \\
Q_3(\alpha) &:= \left(\frac{1-a}{2}\alpha + 1\right)^2.
\end{aligned}
\tag{18}
$$

The following lemma provides the key relation to evaluate the convergence rate of Algorithm 2.

**Lemma 3.5**
*Under Assumptions 1.1 and 2.1, Algorithm 2 generates a sequence $\{x^k\}$ such that*

$$
\frac{\ell R}{2} \geq Q_1(k)u_0(x^{k+1}) + \frac{\ell}{2}Q_2(k)\big\|x^{k+1} - x^k\big\|^2 + \frac{\ell}{2}\sum_{p=1}^{k}P(p)\big\|x^p - x^{p-1}\big\|^2
$$

$$
+ \frac{\ell - L}{2}\sum_{p=1}^{k}Q_3(p)\big\|x^{p+1} - y^{p+1}\big\|^2
$$

*for all $k \geq 0$, where $R \geq 0$ and $P, Q_1, Q_2, Q_3\colon \mathbf{R} \to \mathbf{R}$ are given in (8) and (18), respectively, and $u_0$ is a merit function defined by (5).*

*Proof.* Let $k \geq 0$. With similar arguments used in the proof of Theorem 2.2 (see (Tanabe et al., 2023b, Theorem 5.2)), we get

$$
\sup_{F^* \in F(X^* \cap \mathcal{L}_F(F(x^0)))} \inf_{z \in F^{-1}(\{F^*\})} \sigma_{k+1}(z) = u_0(x^{k+1}).
$$

Since $\rho_k(z) \geq 0$, Lemma 3.4 and the above equality lead to

$$
\begin{aligned}
\frac{\ell R}{2} \geq {} & \frac{1}{1-a}\left[t_{k+1}^2 - at_{k+1} + \left(\frac{1}{4} - b\right)k\right]u_0(x^{k+1}) \\
& + \frac{\ell}{2(1-a)}\left[a(t_{k+1}^2 - t_{k+1}) + \left(\frac{1}{4} - b\right)k\right]\left\|x^{k+1} - x^k\right\|^2 \\
& + \frac{\ell}{2(1-a)}\sum_{p=1}^{k}\left[a^2(t_p - 1) + \left(\frac{1}{4} - b\right)\frac{p - t_p + a(t_p - 1)}{t_p}\right]\left\|x^p - x^{p-1}\right\|^2 \\
& + \frac{\ell - L}{2}\sum_{p=1}^{k}t_{p+1}^2\left\|x^{p+1} - y^{p+1}\right\|^2.
\end{aligned}
$$

We now show that the coefficients of the four terms on the right-hand side can be bounded from below by the polynomials given in (18). First, by using the relation

$$
t_{k+1} \geq \frac{1-a}{2}k + 1 \tag{19}
$$

obtained from Lemma 3.1 (i) and $a \in [0, 1)$, we have

$$
\begin{aligned}
\frac{1}{1-a}\left[t_{k+1}^2 - at_{k+1} + \left(\frac{1}{4} - b\right)k\right] &= \frac{1}{1-a}\left[t_{k+1}(t_{k+1} - a) + \left(\frac{1}{4} - b\right)k\right] \\
&\geq \frac{1}{1-a}\left[\left(\frac{1-a}{2}k + 1\right)\left(\frac{1-a}{2}k + 1 - a\right) + \left(\frac{1}{4} - b\right)k\right] = Q_1(k).
\end{aligned}
$$

Again, (19) gives

$$
\begin{aligned}
\frac{1}{1-a}\left[a(t_{k+1}^2 - t_{k+1}) + \left(\frac{1}{4} - b\right)k\right] &= \frac{a}{1-a}t_{k+1}(t_{k+1} - 1) + \frac{1 - 4b}{4(1-a)}k \\
&\geq \frac{a}{1-a}\left(\frac{1-a}{2}k + 1\right)\left(\frac{1-a}{2}k\right) + \frac{1 - 4b}{4(1-a)}k = Q_2(k).
\end{aligned}
$$

Moreover, since $t_p \leq p$ (cf. Lemma 3.1 (ii)), $t_k \geq 1$ (cf. Lemma 3.1 (i)), and $b \in (a^2/4, 1/4]$, we obtain

$$
\frac{1}{1-a}\left[a^2(t_p - 1) + \left(\frac{1}{4} - b\right)\frac{p - t_p + a(t_p - 1)}{t_p}\right] \geq \frac{a^2}{1-a}(t_p - 1) \geq P(p).
$$

It is also clear from (19) that

$$
t_{p+1}^2 \geq Q_3(p).
$$

Thus, combining the above five inequalities, we get the desired inequality. □

Then, we can finally prove the main theorem.

*Theorem 3.1 (ii).* It is clear from Lemma 3.5 and $Q_1(k) = O(k^2)$ as $k \to \infty$. □

**Remark 3.1**
*Lemma 3.5 also implies the following other claims than Theorem 3.1 (ii):*

- *$O(1/k^2)$ convergence rate of $\left\{\left\|x^{k+1} - x^k\right\|^2\right\}$ when $a > 0$;*

- *the absolute convergence of $\left\{k\left\|x^{k+1} - x^k\right\|^2\right\}$ when $a > 0$;*

- *the absolute convergence of $\left\{k^2\left\|x^k - y^k\right\|^2\right\}$ when $\ell > L$.*

*Note that the second one generalize (Chambolle & Dossal, 2015, Corollary 3.2) for single-objective problems.*

## 4 Convergence of the iterates

While the last section shows that Algorithm 2 has an $O(1/k^2)$ convergence rate like Algorithm 1, this section proves the following theorem, which is more strict than Corollary 2.1 related to Algorithm 1:

**Theorem 4.1**
*Let $\{x^k\}$ be generated by Algorithm 2 with $a > 0$. Then, under Assumptions 1.1 and 2.1, $\{x^k\}$ converges to a weak Pareto optimum for (1).*

This claim is also significant in application. For example, finite-time manifold (active set) identification, which detects the low-dimensional manifold where the optimal solution belongs, essentially requires only the convergence of the generated sequence to a unique point rather than the strong convexity of the objective functions Sun et al. (2019).

Again, we will prove Theorem 4.1 after showing some lemmas. First, we mention the following result, obvious from Assumption 2.1 and Theorem 3.1 (i).

**Lemma 4.1**
*Let $\{x^k\}$ be generated by Algorithm 2 and Assumption 1.1 hold. Then, for any $k \geq 0$, there exists $z \in X^* \cap \mathcal{L}_F(F(x^0))$ (see (3) and (4) for the definitions of $X^*$ and $\mathcal{L}_F$) such that*

$$\sigma_k(z) \geq 0 \quad and \quad \left\| z - x^0 \right\|^2 \leq R,$$

*where $R \geq 0$ is given by (8).*

The following lemma also contributes strongly to the proof of the main theorem.

**Lemma 4.2**
*Let $\{\gamma_q\}$ be defined by line 7 in Algorithm 2. Then, under Assumption 1.1, we have*

$$\sum_{p=s}^{r} \prod_{q=s}^{p} \gamma_q \leq 2(s-1) \quad for\ all\ s, r \geq 1.$$

*Proof.* By using Lemma 3.1 (iv), we see that

$$\prod_{q=s}^{p} \gamma_q \leq \prod_{q=s}^{p} \frac{q-1}{q+1/2}.$$

Let $\Gamma$ and B denote the gamma and beta functions defined by

$$\Gamma(\alpha) := \int_0^{\infty} \tau^{\alpha-1} \exp(-\tau)\,\mathrm{d}\tau \quad and \quad \mathrm{B}(\alpha, \beta) := \int_0^1 \tau^{\alpha-1}(1-\tau)^{\beta-1}\,\mathrm{d}\tau, \tag{20}$$

respectively. Applying the well-known properties:

$$\Gamma(\alpha) = (\alpha-1)!, \quad \Gamma(\alpha+1) = \alpha\Gamma(\alpha), \quad and \quad B(\alpha, \beta) = \frac{\Gamma(\alpha)\Gamma(\beta)}{\Gamma(\alpha+\beta)}. \tag{21}$$

we get

$$\prod_{q=s}^{p} \gamma_q \leq \frac{\Gamma(p)/\Gamma(s-1)}{\Gamma(p+3/2)/\Gamma(s+1/2)} = \frac{B(p, 3/2)}{B(s-1, 3/2)}.$$

This implies

$$\sum_{p=s}^{r} \prod_{q=s}^{p} \gamma_q \leq \sum_{p=1}^{r} B(p, 3/2)/B(s-1, 3/2).$$

Then, it follows from the definition (20) of B that

$$\sum_{p=s}^{r}\prod_{q=s}^{p}\gamma_q \le \sum_{p=s}^{r}\int_0^1 \tau^{p-1}(1-\tau)^{1/2}\,\mathrm{d}\tau/B(s-1,3/2)$$

$$= \int_0^1 \sum_{p=s}^{r}\tau^{p-1}(1-\tau)^{1/2}\,\mathrm{d}\tau/B(s-1,3/2)$$

$$= \int_0^1 \frac{\tau^{s-1}-\tau^r}{1-\tau}(1-\tau)^{1/2}\,\mathrm{d}\tau/B(s-1,3/2)$$

$$= \frac{B(s,1/2)-B(r+1,1/2)}{B(s-1,3/2)} \le \frac{B(s,1/2)}{B(s-1,3/2)}.$$

Using again (21), we conclude that

$$\sum_{p=s}^{r}\prod_{q=s}^{p}\gamma_q \le \frac{\Gamma(s)\Gamma(1/2)/\Gamma(s+1/2)}{\Gamma(s-1)\Gamma(3/2)/\Gamma(s+1/2)} = 2(s-1).$$

$\square$

Now, we introduce two functions $\omega_k\colon \mathbf{R}^n \to \mathbf{R}$ and $\nu_k\colon \mathbf{R}^n \to \mathbf{R}$ for any $k \ge 1$, which will help our analysis, by

$$\omega_k(z) := \max\left(0, \left\|x^k - z\right\|^2 - \left\|x^{k-1} - z\right\|^2\right), \tag{22}$$

$$\nu_k(z) := \left\|x^k - z\right\|^2 - \sum_{s=1}^{k}\omega_s(z). \tag{23}$$

The lemma below describes the properties of $\omega_k$ and $\nu_k$.

**Lemma 4.3**
*Let $\{x^k\}$ be generated by Algorithm 2 and recall that $X^*, \mathcal{L}_F, \omega_k$, and $\nu_k$ are defined by (3), (4), (22) and (23), respectively. Moreover, suppose that Assumptions 1.1 and 2.1 hold and that $z \in X^* \cap \mathcal{L}_F(F(x^0))$ satisfies the statement of Lemma 4.1 for some $k \ge 1$. Then, it follows for all $r = 1,\ldots,k$ that*

*(i)* $\displaystyle\sum_{s=1}^{r}\omega_s(z) \le \sum_{s=1}^{r}(6s-5)\left\|x^s - x^{s-1}\right\|^2$;

*(ii)* $\nu_{r+1}(z) \le \nu_r(z)$.

*Proof.* Claim (i): Let $k \ge p \ge 1$. From the definition of $y^{p+1}$ given in line 8 of Algorithm 2, we have

$$\left\|x^{p+1} - z\right\|^2 - \left\|x^p - z\right\|^2$$
$$= -\left\|x^{p+1} - x^p\right\|^2 + 2\langle x^{p+1} - y^{p+1}, x^{p+1} - z\rangle + 2\gamma_p\langle x^p - x^{p-1}, x^{p+1} - z\rangle$$
$$= -\left\|x^{p+1} - x^p\right\|^2 + 2\langle x^{p+1} - y^{p+1}, y^{p+1} - z\rangle + 2\left\|x^{p+1} - y^{p+1}\right\|^2$$
$$\quad + 2\gamma_p\langle x^p - x^{p-1}, x^{p+1} - z\rangle.$$

On the other hand, Lemma 3.2 (i) gives

$$2\langle x^{p+1} - y^{p+1}, y^{p+1} - z\rangle \le -\frac{2}{\ell}\sigma_{p+1}(z) - \frac{2\ell - L}{\ell}\left\|x^{p+1} - y^{p+1}\right\|^2.$$

Moreover, Lemma 3.3 with $(k_1, k_2) = (p+1, k+1)$ implies

$$-\frac{2}{\ell}\sigma_{p+1}(z) \le -\frac{2}{\ell}\sigma_{k+1}(z) - \left\|x^{k+1} - x^k\right\|^2 + \left\|x^{p+1} - x^p\right\|^2 - \sum_{r=p+1}^{k}\frac{1}{t_r}\left\|x^r - x^{r-1}\right\|^2$$

$$\le \left\|x^{p+1} - x^p\right\|^2,$$

where the second inequality comes from the assumption on $z$. Combining the above three inequalities, we get

$$\left\|x^{p+1} - z\right\|^2 - \left\|x^p - z\right\|^2 \le \frac{L}{\ell}\left\|x^{p+1} - y^{p+1}\right\|^2 + 2\gamma_p\langle x^p - x^{p-1}, x^{p+1} - z\rangle$$

$$= \frac{L}{\ell}\left\|x^{p+1} - y^{p+1}\right\|^2 + \gamma_p\left(\left\|x^p - z\right\|^2 - \left\|x^{p-1} - z\right\|^2 + \left\|x^p - x^{p-1}\right\|^2 + 2\langle x^p - x^{p-1}, x^{p+1} - x^p\rangle\right).$$

Using the relation $\left\|x^{p+1} - y^{p+1}\right\|^2 + 2\gamma_p\langle x^p - x^{p-1}, x^{p+1} - x^p\rangle = \left\|x^{p+1} - x^p\right\|^2 + \gamma_p^2\left\|x^p - x^{p-1}\right\|^2$, which holds from the definition of $y^k$, we have

$$\left\|x^{p+1} - z\right\|^2 - \left\|x^p - z\right\|^2 \le -\frac{\ell - L}{\ell}\left\|x^{p+1} - y^{p+1}\right\|^2 + \left\|x^{p+1} - x^p\right\|^2$$

$$+ \gamma_p\left(\left\|x^p - z\right\|^2 - \left\|x^{p-1} - z\right\|^2\right) + (\gamma_p + \gamma_p^2)\left\|x^p - x^{p-1}\right\|^2.$$

Since $0 \le \gamma_p \le 1$ from Lemma 3.1 (iv) and $\ell \ge L$, we obtain

$$\left\|x^{p+1} - z\right\|^2 - \left\|x^p - z\right\|^2 \le \gamma_p\left(\left\|x^p - z\right\|^2 - \left\|x^{p-1} - z\right\|^2 + 2\left\|x^p - x^{p-1}\right\|^2\right) + \left\|x^{p+1} - x^p\right\|^2$$

$$\le \gamma_p\left(\omega_p(z) + 2\left\|x^p - x^{p-1}\right\|^2\right) + \left\|x^{p+1} - x^p\right\|^2,$$

where the second inequality follows from the definition (22) of $\omega_p$. Since the right-hand side is nonnegative, (22) again gives

$$\omega_{p+1}(z) \le \gamma_p\left(\omega_p(z) + 2\left\|x^p - x^{p-1}\right\|^2\right) + \left\|x^{p+1} - x^p\right\|^2.$$

Let $s \le k$. Applying the above inequality recursively and using $\gamma_1 = 0$, we get

$$\omega_s(z) \le 3\sum_{p=2}^{s}\prod_{q=p}^{s}\gamma_q\left\|x^p - x^{p-1}\right\|^2 + 2\prod_{q=1}^{s}\gamma_q\left\|x^1 - x^0\right\|^2 + \left\|x^s - x^{s-1}\right\|^2$$

$$\le 3\sum_{p=2}^{s}\prod_{q=p}^{s}\gamma_q\left\|x^p - x^{p-1}\right\|^2 + \left\|x^s - x^{s-1}\right\|^2.$$

Adding up the above inequality from $s = 1$ to $s = r \le k$, we have

$$\sum_{s=1}^{r}\omega_s(z) \le 3\sum_{s=1}^{r}\sum_{p=1}^{s}\prod_{q=p}^{s}\gamma_q\left\|x^p - x^{p-1}\right\|^2 + \sum_{s=1}^{r}\left\|x^s - x^{s-1}\right\|^2$$

$$= 3\sum_{p=1}^{r}\sum_{s=p}^{r}\prod_{q=p}^{s}\gamma_q\left\|x^p - x^{p-1}\right\|^2 + \sum_{s=1}^{r}\left\|x^s - x^{s-1}\right\|^2$$

$$= \sum_{s=1}^{r}\left(3\sum_{p=s}^{r}\prod_{q=s}^{p}\gamma_q + 1\right)\left\|x^s - x^{s-1}\right\|^2,$$

where the first equality follows from (14). Thus, Lemma 4.2 implies

$$\sum_{s=1}^{r}\omega_s(z) \le \sum_{s=1}^{r}(6s - 5)\left\|x^s - x^{s-1}\right\|^2.$$

Claim (ii): Equation (23) yields

$$\nu_{r+1}(z) = \left\|x^{r+1} - z\right\|^2 - \omega_{r+1}(z) - \sum_{s=1}^{r} \omega_s(z)$$

$$= \left\|x^{r+1} - z\right\|^2 - \max\left(0, \left\|x^{r+1} - z\right\|^2 - \left\|x^r - z\right\|^2\right) - \sum_{s=1}^{r} \omega_s(z)$$

$$\leq \left\|x^{r+1} - z\right\|^2 - \left(\left\|x^{r+1} - z\right\|^2 - \left\|x^r - z\right\|^2\right) - \sum_{s=1}^{r} \omega_s(z)$$

$$= \left\|x^r - z\right\|^2 - \sum_{s=1}^{r} \omega_s(z) = \nu_r(z),$$

where the second and third equalities come from the definitions (22) and (23) of $\omega_{r+1}$ and $\nu_r$, respectively.
□

Let us now prove the following lemma.

**Lemma 4.4**
*Let $\left\{x^k\right\}$ be generated by Algorithm 2 with $a > 0$. Then, under Assumptions 1.1 and 2.1, $\left\{x^k\right\}$ is bounded, and it has an accumulation point.*

*Proof.* Let $k \geq 1$ and suppose that $z \in X^* \cap \mathcal{L}_F(F(x^0))$ satisfies the statement of Lemma 4.1, where $X^*$ and $\mathcal{L}_F$ are given by (3) and (4), respectively. Then, Lemma 4.3 (ii) gives

$$\nu_k(z) \leq \nu_1(z) = \left\|x^1 - z\right\|^2 - \omega_1(z)$$

$$= \left\|x^1 - z\right\|^2 - \max\left(0, \left\|x^1 - z\right\|^2 - \left\|x^0 - z\right\|^2\right)$$

$$\leq \left\|x^1 - z\right\|^2 - \left(\left\|x^1 - z\right\|^2 - \left\|x^0 - z\right\|^2\right) = \left\|x^0 - z\right\|^2,$$

where the second equality follows from the definition (22) of $\omega_1$. Considering the definition (23) of $\nu_k$, we obtain

$$\left\|x^k - z\right\|^2 \leq \left\|x^0 - z\right\|^2 + \sum_{s=1}^{k} \omega_s(z).$$

Taking the square root of both sides and using (22), we get

$$\left\|x^k - z\right\| \leq \sqrt{\left\|x^0 - z\right\|^2 + \sum_{s=1}^{k} (6s - 5)\|x^s - x^{s-1}\|^2}.$$

Applying the reverse triangle inequality $\left\|x^k - x^0\right\| - \left\|x^0 - z\right\| \leq \left\|x^k - z\right\|$ to the left-hand side leads to

$$\left\|x^k - x^0\right\| \leq \left\|x^0 - z\right\| + \sqrt{\left\|x^0 - z\right\|^2 + \sum_{s=1}^{k} (6s - 5)\|x^s - x^{s-1}\|^2}$$

$$\leq \sqrt{R} + \sqrt{R + \sum_{s=1}^{k} (6s - 5)\|x^s - x^{s-1}\|^2},$$

where the second inequality comes from the assumption on $z$. Moreover, since $a > 0$, the right-hand side is bounded from above according to Lemma 3.5. This implies that $\left\{x^k\right\}$ is bounded, and so it has accumulation points. □

Before proving Theorem 4.1, we show the following lemma.

**Lemma 4.5**
*Let $\{x^k\}$ be generated by [Algorithm 2](#) with $a > 0$ and suppose that [Assumptions 1.1](#) and [2.1](#) holds. Then, if $\bar{z}$ is an accumulation point of $\{x^k\}$, then $\{\|x^k - \bar{z}\|\}$ is convergent.*

*Proof.* Assume that $\{x^{k_j}\} \subseteq \{x^k\}$ converges to $\bar{z}$. Then, we have $\sigma_{k_j}(\bar{z}) \to 0$ by the definition [(12)](#) of $\sigma_{k_j}$. Therefore, we can regard $\bar{z}$ to satisfy the statement of [Lemma 4.1](#) at $k = \infty$, and thus the inequalities of [Lemma 4.3](#) hold for any $r \geq 1$ and $\bar{z}$. This means $\{\nu_k(\bar{z})\}$ is non-increasing and bounded, i.e., convergent. Hence $\{\|x^k - \bar{z}\|\}$ is convergent. $\qquad\square$

Finally, we finish the proof of the main theorem.

*Proof of [Theorem 4.1](#).* Suppose that $\left\{x^{k_j^1}\right\}$ and $\left\{x^{k_j^2}\right\}$ converges to $\bar{z}^1$ and $\bar{z}^2$, respectively. From [Lemma 4.5](#), we see that

$$\lim_{j \to \infty} \left( \left\|x^{k_j^2} - \bar{z}^1\right\|^2 - \left\|x^{k_j^2} - \bar{z}^2\right\|^2 \right) = \lim_{j \to \infty} \left( \left\|x^{k_j^1} - \bar{z}^1\right\|^2 - \left\|x^{k_j^1} - \bar{z}^2\right\|^2 \right).$$

This yields that $\left\|\bar{z}^1 - \bar{z}^2\right\|^2 = -\left\|\bar{z}^1 - \bar{z}^2\right\|^2$, and so $\left\|\bar{z}^1 - \bar{z}^2\right\|^2 = 0$, i.e., $\{x^k\}$ is convergent. Let $x^k \to x^*$. Since $\left\|x^{k+1} - x^k\right\|^2 \to 0$, $\{y^k\}$ is also convergent to $x^*$. Therefore, [Proposition 2.1](#) shows that $x^*$ is weakly Pareto optimal for [(1)](#). $\qquad\square$

# 5 Numerical experiments

This section compares the performance between [Algorithm 2](#) with various $a$ and $b$ and [Algorithm 1](#) ($a = 0, b = 1/4$) through numerical experiments. Our newly introduced generalized momentum factor, while not primarily focused on improving convergence rates, serves to provide a theoretical link between different accelerated gradient methods. The primary goal of the numerical experiments is to confirm that our proposed method performs as theoretically expected. At the same time, it suggests that some momentum factors may potentially lead to better results. We run all experiments in Python 3.9.9 on a machine with 2.3 GHz Intel Core i7 CPU and 32 GB memory. For each example, we test 15 different hyperparameters combining $a = 0, 1/6, 1/4, 1/2, 3/4$ and $b = a^2/4, (a^2+1)/8, 1/4$, i.e.,

$$(a, b) = \left\{ \begin{array}{c} (0, 0), (0, 1/8), (0, 1/4), \\ (1/6, 1/144), (1/6, 37/288), (1/6, 1/4), \\ (1/4, 1/64), (1/4, 17/128), (1/4, 1/4), \\ (1/2, 1/16), (1/2, 5/32), (1/2, 1/4), \\ (3/4, 9/64), (3/4, 25/128), (3/4, 1/4) \end{array} \right\},$$

and we set $\varepsilon = 10^{-5}$ for the stopping criteria. The source code we used is available as open source at [https://github.com/zalgo3/zfista](https://github.com/zalgo3/zfista).

## 5.1 Artificial test problems (bi-objective and tri-objective)

First, we focus on solving multi-objective test problems, which are generally formulated as in problem [(1)](#). Specifically, we use part of the test problems of [Tanabe et al. (2023b)](#), whose objective functions are defined

by

$$f_1(x) = \frac{1}{n}\|x\|^2, f_2(x) = \frac{1}{n}\|x - 2\|^2, g_1(x) = g_2(x) = 0, \tag{JOS1}$$

$$f_1(x) = \frac{1}{n}\|x\|^2, f_2(x) = \frac{1}{n}\|x - 2\|^2, g_1(x) = \frac{1}{n}\|x\|_1, g_2(x) = \frac{1}{2n}\|x - 1\|_1, \tag{JOS1-L1}$$

$$\begin{cases} f_1(x) = \frac{1}{n^2}\sum_{i=1}^{n} i(x_i - i)^4, f_2(x) = \exp\left(\sum_{i=1}^{n}\frac{x_i}{n}\right) + \|x\|^2, \\[3mm] f_3(x) = \frac{1}{n(n+1)}\sum_{i=1}^{n} i(n - i + 1)\exp(-x_i), g_1(x) = g_2(x) = g_3(x) = 0, \end{cases} \tag{FDS}$$

$$\begin{cases} f_1(x) = \frac{1}{n^2}\sum_{i=1}^{n} i(x_i - i)^4, f_2(x) = \exp\left(\sum_{i=1}^{n}\frac{x_i}{n}\right) + \|x\|^2, \\[3mm] f_3(x) = \frac{1}{n(n+1)}\sum_{i=1}^{n} i(n - i + 1)\exp(-x_i), g_1(x) = g_2(x) = g_3(x) = \chi_{\mathbf{R}_+^n}(x), \end{cases} \tag{FDS-CON}$$

where $x \in \mathbf{R}^n, n = 50$ and $\chi_{\mathbf{R}_+^n}$ is an indicator function (2) of the nonnegative orthant. These problems include modifications inspired by Jin et al. (2001); Fliege et al. (2009). We have chosen the problems because they cover bi-objective and tri-objective problems with non-differentiable or constrained cases. While Tanabe et al. (2023b) covers more problems, we have narrowed down the problems to avoid complicating the publication of the results since this numerical experiment involves numerous problems with different $(a, b)$ problems. The solver is open source and can be used by anyone, so readers interested in results for other problems are welcome to follow up.

We choose 1000 initial points, commonly for all pairs $(a, b)$, and randomly with a uniform distribution between $\underline{c}$ and $\overline{c}$, where $\underline{c} = (-2, \ldots, -2)^\top$ and $\overline{c} = (4, \ldots, 4)^\top$ for (JOS1) and (JOS1-L1), $\underline{c} = (-2, \ldots, -2)^\top$ and $\overline{c} = (2, \ldots, 2)^\top$ for (FDS), and $\underline{c} = (0, \ldots, 0)^\top$ and $\overline{c} = (2, \ldots, 2)^\top$ for (FDS-CON). Moreover, we use backtracking for updating $\ell$, with 1 as the initial value of $\ell$ and 2 as the constant multiplied into $\ell$ at each iteration (cf. (Tanabe et al., 2023b, Remark 4.1 (v))). Furthermore, at each iteration, we transform the subproblem (6) into their dual as suggested in Tanabe et al. (2023b) and solve them with the trust-region interior point method Byrd et al. (1999) using the scientific library SciPy.

Figure 1 and Table 1 present the experimental results. Figure 1 plots the objective function values at the points where the stopping criteria is satisfied for each problem. We only show the cases $(a, b) = (0, 1/4), (3/4, 1/4)$, but other combinations also yield similar plots, including a wide range of Pareto solutions. Table 1 lists the average time and average number of iterations until satisfying the stopping criteria for each initial point, for each problem, and for each $a, b$. This shows that the new momentum factors are fast enough to compete with the existing ones $((a, b) = (0, 1/4)$ or $b = a^2/4)$ and better than them in some cases.

## 5.2  Image deblurring (single-objective)

Since our proposed momentum factor is also new in the single-objective context, we also tackle deblurring the cameraman test image via a single-objective $\ell_2$-$\ell_1$ minimization, inspired by Beck & Teboulle (2009). This experiment also aims to show that our momentum coefficients, which combine existing well-known momentum coefficients while ensuring convergence of the point sequence, perform comparably well for application tasks. Several methods for $\ell_2$-$\ell_1$ minimization are known, such as ISTA and TWIST Bioucas-Dias & Figueiredo (2007), but comparisons between them and FISTA have already been made in Beck & Teboulle (2009) and others. Therefore, in this paper we only compare FISTA with the proposed new momentum coefficients. In detail, as shown in Figure 2, to a $256 \times 256$ cameraman test image with each pixel scaled to $[0, 1]$, we generate an observed image by applying a Gaussian blur of size $9 \times 9$ and standard deviation 4 and adding a zero-mean white Gaussian noise with standard deviation $10^{-3}$.

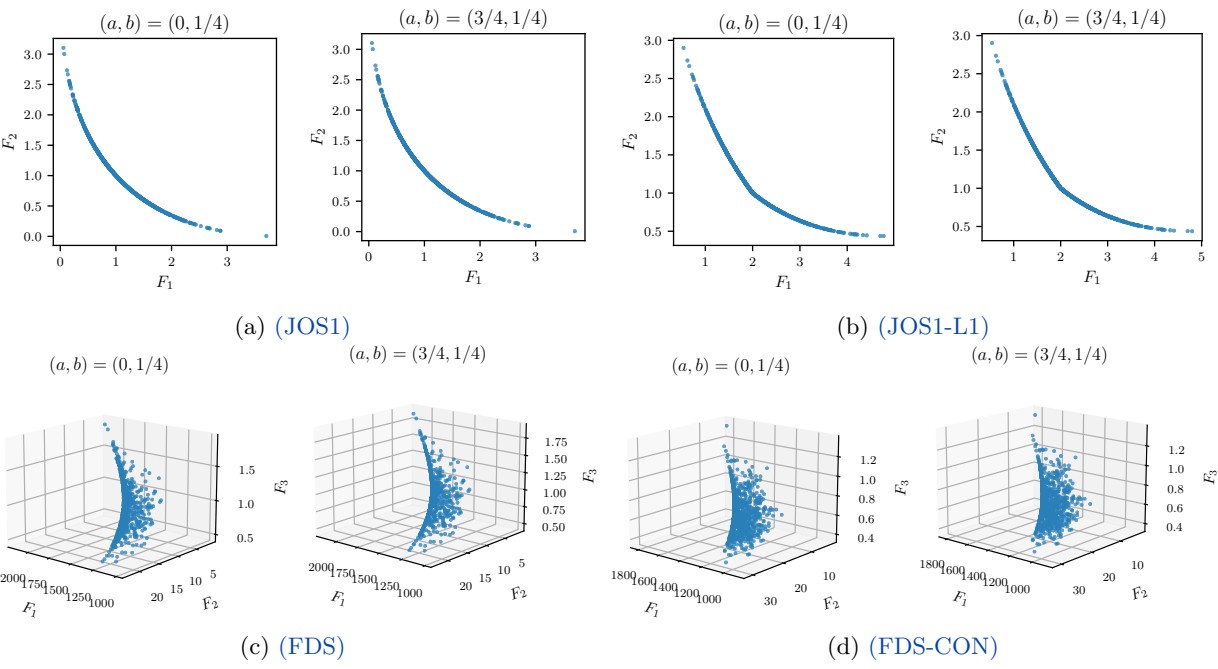

(a) (JOS1)

(b) (JOS1-L1)

(c) (FDS)

(d) (FDS-CON)

Figure 1: Pareto solutions obtained with some $(a, b)$

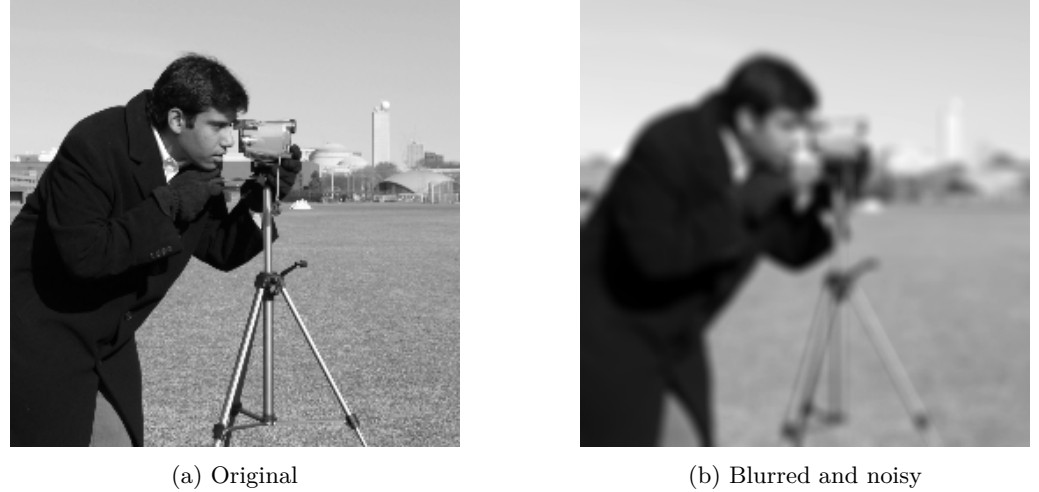

(a) Original

(b) Blurred and noisy

Figure 2: Deblurring of the cameraman

Table 1: Average computational costs to solve the multi-objective examples

(a) (JOS1)

| $a$ | $b$ | Time [s] | Iterations |
|---|---|---|---|
| 0 | 0 | 6.442 | 97.0 |
| 0 | 1/8 | 5.158 | 81.217 |
| 0 | 1/4 | 4.207 | 65.0 |
| 1/6 | 1/144 | 4.244 | 67.0 |
| 1/6 | 37/288 | 5.182 | 82.0 |
| 1/6 | 1/4 | 4.268 | 66.0 |
| 1/4 | 1/64 | 6.224 | 99.0 |
| 1/4 | 17/128 | 7.239 | 113.566 |
| 1/4 | 1/4 | 3.205 | 51.0 |
| 1/2 | 1/16 | 4.51 | 72.0 |
| 1/2 | 5/32 | 4.562 | 71.0 |
| 1/2 | 1/4 | 4.466 | 70.0 |
| 3/4 | 9/64 | 4.323 | 67.998 |
| 3/4 | 25/128 | 3.104 | 49.0 |
| 3/4 | 1/4 | 3.741 | 47.0 |

(b) (JOS1-L1)

| $a$ | $b$ | Time [s] | Iterations |
|---|---|---|---|
| 0 | 0 | 10.733 | 157.512 |
| 0 | 1/8 | 11.054 | 161.065 |
| 0 | 1/4 | 11.122 | 161.734 |
| 1/6 | 1/144 | 9.85 | 141.731 |
| 1/6 | 37/288 | 9.994 | 144.863 |
| 1/6 | 1/4 | 10.399 | 150.592 |
| 1/4 | 1/64 | 9.271 | 135.804 |
| 1/4 | 17/128 | 9.463 | 137.108 |
| 1/4 | 1/4 | 9.662 | 139.848 |
| 1/2 | 1/16 | 7.439 | 109.082 |
| 1/2 | 5/32 | 7.642 | 110.204 |
| 1/2 | 1/4 | 7.723 | 111.599 |
| 3/4 | 9/64 | 5.253 | 77.366 |
| 3/4 | 25/128 | 5.39 | 79.425 |
| 3/4 | 1/4 | 5.678 | 82.37 |

(c) (FDS)

| $a$ | $b$ | Time [s] | Iterations |
|---|---|---|---|
| 0 | 0 | 29.24 | 204.438 |
| 0 | 1/8 | 29.797 | 210.595 |
| 0 | 1/4 | 30.565 | 214.934 |
| 1/6 | 1/144 | 24.964 | 174.393 |
| 1/6 | 37/288 | 25.375 | 177.944 |
| 1/6 | 1/4 | 26.065 | 182.398 |
| 1/4 | 1/64 | 22.94 | 159.737 |
| 1/4 | 17/128 | 23.311 | 162.629 |
| 1/4 | 1/4 | 23.976 | 166.918 |
| 1/2 | 1/16 | 17.909 | 122.653 |
| 1/2 | 5/32 | 18.14 | 123.96 |
| 1/2 | 1/4 | 18.221 | 125.697 |
| 3/4 | 9/64 | 13.584 | 94.176 |
| 3/4 | 25/128 | 13.674 | 94.705 |
| 3/4 | 1/4 | 13.795 | 94.868 |

(d) (FDS-CON)

| $a$ | $b$ | Time [s] | Iterations |
|---|---|---|---|
| 0 | 0 | 37.345 | 259.508 |
| 0 | 1/8 | 37.439 | 261.522 |
| 0 | 1/4 | 37.94 | 263.911 |
| 1/6 | 1/144 | 32.463 | 227.063 |
| 1/6 | 37/288 | 38.265 | 229.736 |
| 1/6 | 1/4 | 45.661 | 231.958 |
| 1/4 | 1/64 | 41.434 | 209.35 |
| 1/4 | 17/128 | 33.664 | 211.69 |
| 1/4 | 1/4 | 30.772 | 213.811 |
| 1/2 | 1/16 | 22.92 | 158.448 |
| 1/2 | 5/32 | 23.1 | 159.685 |
| 1/2 | 1/4 | 23.539 | 162.226 |
| 3/4 | 9/64 | 17.092 | 118.616 |
| 3/4 | 25/128 | 17.123 | 118.063 |
| 3/4 | 1/4 | 17.115 | 118.844 |

Letting $\theta, B$, and $W$ be the observed image, the blur matrix, and the inverse of the Haar wavelet transform Haar (1910), respectively, consider the single-objective problem (1) with $m = 1$ and

$$f_1(x) := \|BWx - \theta\|^2 \quad \text{and} \quad g_1(x) = \lambda\|x\|_1,$$

where $\lambda := 2 \times 10^{-5}$ is a regularization parameter. Unlike in the previous subsection, we can compute $\nabla f$'s Lipschitz constant by calculating $(BW)^\top(BW)$'s eigenvalues using the two-dimensional cosine transform Hansen et al. (2006), so we use it constantly as $\ell$. Moreover, we use the observed image's Wavelet transform as the initial point.

Figure 3 shows the reconstructed image from the obtained solution. Although there are some quirks in the way images are deblurred, such as the way stripes remain depending on the hyperparameters, it can be observed that deblurring is generally successful for all parameters. Moreover, we summarize the numerical performance in Table 2: each row represents the performance for each $(a, b)$, and the columns "Time [s]" and "Iteration counts" are the time and the number of iterations until the termination condition is met, respectively, and the column "$F_1(x^{200})$" represents the objective function value at iteration 200. Like the last subsection, this example also suggests that some of our new momentum factors may occasionally improve the algorithm's performance even for single-objective problems.

Table 2: Computational costs for the image deblurring

| $a$ | $b$ | Total time [s] | Iteration counts | $F_1(x^{200})$ |
|-----|-----|----------------|------------------|----------------|
| 0 | 0 | 85.391 | 517 | 10.285 |
| 0 | 1/8 | 85.037 | 517 | 10.367 |
| 0 | 1/4 | 85.128 | 517 | 10.456 |
| 1/6 | 1/144 | 80.692 | 480 | 8.867 |
| 1/6 | 37/288 | 80.833 | 480 | 8.88 |
| 1/6 | 1/4 | 81.449 | 480 | 8.904 |
| 1/4 | 1/64 | 71.583 | 417 | 8.491 |
| 1/4 | 17/128 | 71.165 | 417 | 8.459 |
| 1/4 | 1/4 | 48.997 | 416 | 8.442 |
| 1/2 | 1/16 | 39.447 | 319 | 9.63 |
| 1/2 | 5/32 | 41.76 | 318 | 9.351 |
| 1/2 | 1/4 | 41.122 | 318 | 9.125 |
| 3/4 | 9/64 | 47.621 | 399 | 23.558 |
| 3/4 | 25/128 | 43.671 | 393 | 21.832 |
| 3/4 | 1/4 | 40.17 | 388 | 20.493 |

## 6 Conclusion

We have generalized the momentum factor of the multi-objective accelerated proximal gradient algorithm Tanabe et al. (2023b) in a form that is even new in the single-objective context and includes the known FISTA momentum factors Beck & Teboulle (2009); Chambolle & Dossal (2015). Furthermore, with the proposed momentum factor, we proved under reasonable assumptions that the algorithm has an $O(1/k^2)$ convergence rate and that the iterates converge to Pareto solutions. To the best of our knowledge, the proposed method is the first to demonstrate convergence of the iterates with the accelerated gradient method for multi-objective optimization problems. Moreover, the numerical results reinforced these theoretical properties and suggested the potential for our new momentum factor to improve the performance. In practical operation, hyperparameter tuning with our momentum factor for each type of task may lead to faster solutions than conventional algorithms.

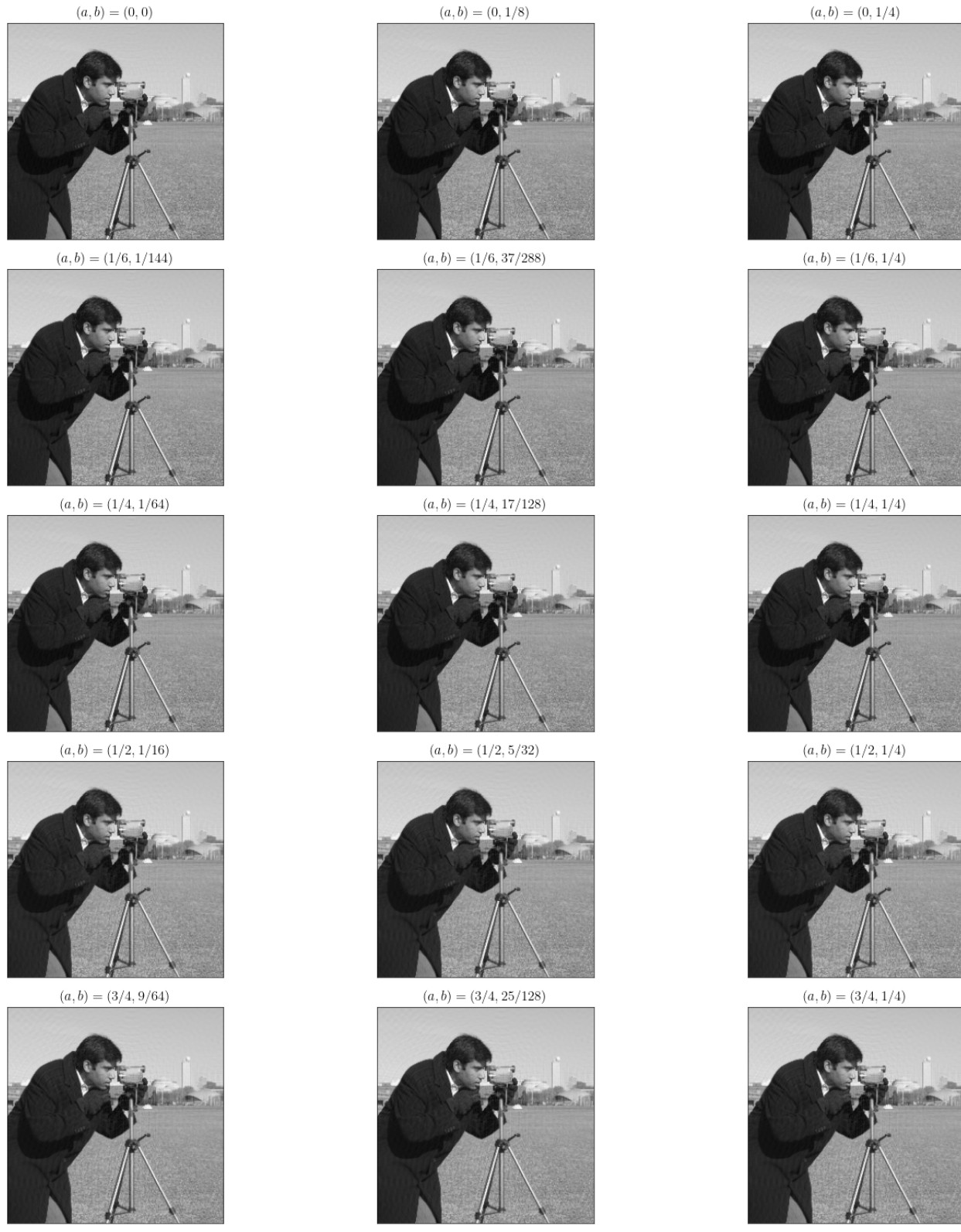

Figure 3: Deblurred image

As we mentioned in Section 4, our proposed method has the potential to achieve finite-time manifold (active set) identification Sun et al. (2019) without the assumption of the strong convexity (or its generalizations such as PL conditions or error bounds Karimi et al. (2016)). Moreover, we took a single update rule of $t_k$ for all iterations in this work, but the adaptive change of the strategy in each iteration is conceivable. It might also be interesting to estimate the Lipschitz constant simultaneously with that change, like in Scheinberg et al. (2014). In addition, an extension to the inexact scheme like Villa et al. (2013) would be significant. Furthermore, in single-objective optimization, non-convex objectives for FISTA have been proposed Li & Lin (2015), and extending this approach to multi-objective optimization remains an open problem. Regarding the application of our method in settings where only stochastic gradients are available, adapting our approach to such scenarios is an interesting direction for future research. Recent studies in multi-objective optimization with stochastic gradients Liu & Vicente (2021); Zhou et al. (2022) provide valuable insights and foundations for such an adaptation. This is an open area for exploration, possibly in conjunction with techniques such as dual averaging Xiao (2010). Those are issues to be addressed in the future.

## Acknowledgements

This work was supported by the Grant-in-Aid for Scientific Research (C) (21K11769 and 19K11840) and Grant-in-Aid for JSPS Fellows (20J21961) from the Japan Society for the Promotion of Science.

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
