# OpenReview forum: "A globally convergent fast iterative shrinkage-thresholding algorithm with a new momentum factor for single and multi-objective convex optimization"
_TMLR — Rejected by TMLR_

### Review · Reviewer_Dmn6 · 2024-09-25

**Summary Of Contributions:**

The authors propose an extension of an accelerated algorithm to find weak Pareto solution of multi-objective minimization of convex functions. The main contribution is to modify the momentum parameter by adding two parameters $a,\, b$ as follow:
$$
t_{k+1} = \sqrt{t_k^2 - a t_k + b} + 1/2,\qquad \gamma_k = (t_k-1)/t_{k+1}
$$
instead of the classical algorithm where $a = 0$ and $b = 1/4$. The authors show that by setting $0<a<1$, $a^2/4 \leq b  \leq 1/4$, it is possible to show the convergence of the iterates toward a Pareto optimal point.

The authors also analyze numerically the effect of $a, \, b$ on several problems.

**Audience:**

Yes

**Broader Impact Concerns:**

N/A, the work is purely theoretical.

**Claims And Evidence:**

Yes

**Requested Changes:**

- The output of Algorithm 1 and 2 is unclear:
1) What iterate is the output? I suppose it is $x_k$, but this should be clarified.
2) The output is not $x^*$, a weakly Pareto optimal point, but rather an approximation. Moreover. the approximation term is not specified in the algorithm (e.g. a point that satisfies $u(x) < \epsilon$).

- The first equation, $\min F(x)$, is not well defined, as we cannot minimize vector. I understand the idea is to emphasize the multi-objective problem, but writing that the goal is to minimize a vector is mathematically wrong. The best would be to immediately introduce the goal of finding a weak Pareto optimal point.

Also, please check the section "Strengths And Weaknesses"

**Strengths And Weaknesses:**

# Strength

- Clarity: Overall, the paper is well written and the proof sufficiently detailed to check all steps, which is appreciated.
- Novelty / literature review: The authors did a good job comparing their contribution w.r.t previous, similar work, and exiting results are always cited. Moreover, according to the literature review, the paper explores a new idea which was not studied before.

# Weaknesses

There are a few point that, I think, are problematic in the paper. The point are listed in order of importance, starting with most critical one.

## Unclear advantage of finding a Pareto point VS minimizing a function

The paper interest is to find a weak Pareto solution to the problem
$$ \min_x F(x) = [F_1(x), \ldots, F_m(x) ] $$
where $F(x)$ is a vector valuated function. The authors uses the definition of weak Pareto point $x^*$ to be
$$ x^* : \not \exists y : F_i (y) < F_i(x) \quad \forall i=1\ldots m $$
i.e., it is impossible to find another point $y$ such that $y$ strictly decreases all function value.

It happens that finding any minimizers of $F_i(x)$ (i.e., focusing on a specific, arbitrary function) will always lead to a weak pareto solution, since by definition, it is impossible to find another point that decreases strictly that function more. This means that finding a weak Pareto point should be *easier* than finding the minimum of a function, since the minimum of a specific function is a particular case.

This specific point made me question what is the interest of finding weak Pareto point VS minimizing just an arbitrary function (i.e. picking an arbitrary index $i$ and minimize $F_i(x)$)? I believe the authors should empathize more on the difference / on why minimizing just an arbitrary function is not desirable.


## Misleading link between the merit function VS what is actually minimized / Why choosing this specific merit function?

The merit function in the paper is defined as
$$ u(x) = \max_z \min_i F_i(x) - F_i(z) $$
and $u(x) = 0$ if and only if $x$ is Pareto optimal. However, the algorithm presented in the paper actually minimize another function. Let us call
$$ h(x,y) = \max_i F_i(x) - F_i(z). $$

For all $y$, the function $h(x, y)$ is a convex function since $F_i$ is convex too. Then, the algorithm is actually making one iteration on the function
$$ h(x, x_{k-1}) $$
i.e., it minimize $h(x, \cdot)$ w.r.t. $x$.

It was quite hard to understand, at first, that the approach was *not* to minimize the merit function (which was non convex), but rather to minimize the function $h$ I introduced earlier.

Moreover, while I understand that the merit function $u(x)$ is convenient to measure the optimality of $x$ (i.e., $=0$ iff $x$ is Pareto optimal), the authors do not justify what are the advantages of this merit function when it comes to the algorithm design, especially given the fact that they are not minimizing it directly. Also, since the goal is to find one weak Pareto optimal point, the "if and only if" is not required - in fact, any merit function $\tilde u$ such that $\tilde{u}(x^*) = 0 \Rightarrow x^*$ is pareto optimal is sufficient for the algorithm design.

Morevoer, I would like to know what would be the advantage oif the proposed method VS this one:

> Having several x_i that minimizes each F_i, then stop when one of those x_i reaches the minimum

Note that this has the same gradient oracle call complexity per iteration, and also converges in $O(1/T^2)$ if we use accelerated gradient descent.

## Limitation: Potential evaluation outside domf

The algorithm requires to evaluate the function at $x_{k-1}$ and $y_k$ to compute the next iterates. However, by construction, even if $x\in \text{domf}$, this might not be the case for $y$ since it perform a linear (but not convex) combination of $x_{k} and $x_{k-1}, which might land outside of domf.


## Unclear consequence of changing $a$ and $b$

While exploring numerically the impact of changing $a$ and $b$, there are no guideline / theoretical study of the impact of changing / optimal value of the parameters $a$ and $b$.

---

### Review · Reviewer_HM3J · 2024-10-28

**Summary Of Contributions:**

This paper extends the theoretical study of accelerated proximal gradient method for solving multi-objective convex-composite optimization. The paper generalizes hyperparameters (a, b) associated with momentum factor in the accelerated proximal gradient method. Rigorous convergence analysis is conducted to the same convergence rate $O(1/k^2)$ as the previous work of Tanabe et al. (2023b) in terms of optimality gap.  The authors also prove the convergence to Pareto optimal points of the whole iterate sequence. The generalized momentum factor provides a theoretical link between different accelerated gradient methods. Their numerical results demonstrate the potential for generalized momentum factors to improve the convergence performance.

**Audience:**

No

**Claims And Evidence:**

Yes

**Requested Changes:**

-  Theorem 4.1 discussed in Section 4 is novel and main adds-on to the existing theoretical framework. More clarification about it is important.
    - Given that (ii) implies (i), what’s the purpose of presenting them as separate properties?
    - In the proof of Theorem 4.1 (ii), it is not clear to me how you derive the equality of the two lim terms from Lemma 4.4?
- One missing piece is how the convexity impacts the convergence rate. The problem formulation (1) states that both component functions are convex. However, all the theorems rely on Assumption 2.1 where convexity is not required explicitly. Could you elaborate more on how the convexity is used in the proof?

**Strengths And Weaknesses:**

The paper is well-organized and easy to follow. The theoretical study is well-structured and the provided numerical experiments are extensive enough to confirm the algorithm performance. However, the scope and novelty of this work are limited. It’s essentially an extension of the previous work of Tanabe et al. (2023b), i.e., the algorithm assumption and the first main theorem’s proof logic are inherited from them. The numerical experiment demonstrates similar performance of different hyper-parameter combinations to fixed ones commonly adopted in the literature. It is not clear to the reader: (1) what the gain from tuning this additional set of hyper-parameters is;  (2) how this method advances ML real applications with either multi-objective or single-objective optimization.

---

### Review · Reviewer_APsx · 2024-10-30

**Summary Of Contributions:**

For multi-objective convex-composite optimization, existing FISTA has classical momentum, and has gap in proving $O(1/k^2)$ rate. This work proposes another APG with general momentum, shows $O(1/k^2)$ rate, and shows convergence to a weak Pareto solution, with extension to finite-time manifold identification: achieving finite-iteeration manifold identification without assuming strong convexity. The general momentum parameters are novel, are proven with optimal rate, and performs well. Solid novel theoretical analysis with proofs are established. Numerical results on synthetic (tri-obj) test and image deblurring (single-obj) are presented.

**Audience:**

Yes

**Claims And Evidence:**

Yes

**Requested Changes:**

In general, please try to address all weaknesses about numerical section.
1. For both numerical experiments, please sufficiently explain all formulations, tables, figures, summarized results.
2. Adding another multi-objective real experiment is desired.

**Strengths And Weaknesses:**

Strengths:
1. On multi-obj convex-composite problem, the proposed method is able to reach $O(1/k^2)$ rate, can converge to a weak Pareto solution, with extension to finite-time manifold identification. The general momentum parameters are novel, are proven with optimal rate, and performs well. The main theoretical result is good. The theoretical analysis is in general solid, novel, and interesting.

Weaknesses:
1. For the numerical setting in general, one synthetic multi-objective problem + one single-objective image deblurring is very insufficient.
2. Numerical section is way too vague. For example, Many Tables and Figures in numerical section are just provided with almost no explanation. Formulation for problem 1 as well as both problems’ results are also almost not explained at all.
3. The formulation of numerical experiment 1 (artificial tests) is not explained at all besides the few references provided. Also, among multi-objective convex problems, not clear why this particular artificial problem is a good problem to evaluate, and whether it has any practical application.
4. The only real experiment (Exp.2 image deblurring) is single-objective. and has the simplest math formulation.
5. According to results presented in the 2 tables, the proposed method does not improve much over existing ones. The potential advantage is only occasional, as the author points out as well.

---

### Review · Reviewer_vxhb · 2024-10-30

**Summary Of Contributions:**

The authors generalize the momentum of the step size in the accelerated gradient method for multi-objective convex optimization. The authors prove that the proposed method can converge with the rate $\mathcal{O}(1/k^2)$ which is the best rate in the literature. The proposed method is validated on synthesis function and image debluring.

**Audience:**

Yes

**Claims And Evidence:**

Yes

**Requested Changes:**

1. All assumptions used in the Theorem should be clearly stated.

2. Additional comparison of image deblurring.

**Strengths And Weaknesses:**

## Strength:

1. The authors propose a more general framework of the step size that includes serval methods (e.g. accelerated proximal gradient descent)

2.  The authors prove the proposed method converges in the order $\mathcal{O}(1/k^2)$, which achieves the optimal rate for the first-order method for convex optimization.

## Weaknesses:

1.  Assumptions should be clearly stated. function $f,g$ are convex, $f$ has Lipschitz gradients, $l \geq L$, should be clearly stated in the Theorem or assumptions.

2.  Does $\{x^k\}$ have convergence rate? If not, can the author briefly discuss, why one can show the convergence rate for the "function u(x)" but can not give convergence rate for $x$ in convex optimization?

Meanwhile, since $a>0$ is the condition of Theorem 4.1, does this mean the proof can not cover the Nesterov momentum?

3. For the image deblurring, other than the type of proposed algorithm, the authors should compare it with other optimization algorithms of solving this LASSO problem.

---

### Decision · Action_Editor_JyZc · 2025-01-03

**Recommendation:** Reject

**Comment:**

This submission concerns multi-objective composite optimization. In particular, the problem is minimizing $F(x)$ where $F = (F_1, \dots, F_m)^\top$ and $F_i(x) = f_i(x) + g_i(x)$ with $f_i \colon\mathbb{R}^n\to \mathbb{R}$ being convex and smooth, and $g_i\colon\mathbb{R}^n \to \mathbb{R} \cup\{+\infty\}$ being proper, convex, closed with an efficiently computable proximal operator. When $m=1$, this problem reduces to classical (also referred to as the single-objective) composite optimization. The aim of the submission is to design an accelerated proximal gradient method for the multi-objective template with $O(1/k^2)$ convergence rate (for a merit function corresponding to finding an approximately weakly Pareto optimal point) and convergence of the sequence to a weakly Pareto optimal point.

The submission received four reviews. The most major issue, as clearly delineated by Reviewer Dmn6, is the notion of weak Pareto optimality adopted in the submission. In particular, Reviewer Dmn6 pointed out that this can be a quite weak measure and this is in fact also acknowledged in the classical literature for multi-objective optimization (see also Theorem 3.1.1 in Miettinen's book cited in the submission). For example, [Section 4, I] and [Section 2.3, II] state this issue and [Section 4, I] also include ways to strengthen their results to obtain Pareto optimal points, instead of weakly Pareto optimal points.

As a result, the authors are expected to either strengthen their results, by for example using ideas from [Section 4, I] and related references; or to address the concerns of Reviewer Dmn6 and justify theoretically how the weakly Pareto optimal points generated by their algorithms are favorable for multi-objective problems (compared to other ways of generating weakly Pareto points pointed out by Reviewer Dmn6). This is important since as the authors argue, the contribution of the submission is theoretical.

The authors are also recommended to not share github repository links that may violate anonymity (such as the one they have right now in their rebuttal and their main text).

[I] Bonnel, Henri, Alfredo Noel Iusem, and Benar Fux Svaiter. "Proximal methods in vector optimization." SIAM Journal on Optimization 15, no. 4 (2005): 953-970.

[II] Jahn, Johannes. "Theory of vector maximization: Various concepts of efficient solutions." In Multicriteria decision making: Advances in MCDM models, algorithms, theory, and applications, pp. 37-68. Boston, MA: Springer US, 1999.

**Audience:**

The reviewing team agrees that the submission could be of interest to some individuals in TMLR's audience.

**Claims And Evidence:**

The reviewing team concluded that the claims in the submission, particularly solving multi-objective optimization problems, are not clearly supported by evidence due to issues with the notion of weak Pareto optimality used in the theoretical results. More details are given below in the "Comment" section.

**Resubmission Of Major Revision:**

The authors may consider submitting a major revision at a later time.